# What are the benefits of lidar-assisted control in the design of a wind turbine?

Helena Canet[1], Stefan Loew[1], and Carlo L. Bottasso[1]

[1]Wind Energy Institute, Technical University of Munich, 85748 Garching b. München, Germany

**Correspondence:** Carlo L. Bottasso (carlo.bottasso@tum.de)

**Abstract.**

This paper explores the potential benefits brought by the integration of lidar-assisted control (LAC) in the design of a wind turbine. The study identifies which design drivers can be relaxed by LAC, and by how much these drivers could be reduced before other conditions become the drivers. A generic LAC load-reduction model is defined and used to redesign the rotor and tower of three representative turbines, differing in terms of wind class, size and power rating. The load reductions enabled by LAC are used to save mass, increase hub height or extend lifetime. For the first two strategies, results suggest only modest reductions in the levelized cost of energy, with potential benefits essentially limited to the tower of a large offshore machine. On the other hand, lifetime extension appears to be the most effective way of exploiting the effects of LAC.

## 1  Introduction

Wind turbines are highly dynamical systems, excited by stochastic and deterministic disturbances from wind. Among their various goals, wind turbine control systems aim at limiting structural loads. In fact, lower ultimate and fatigue loading can be exploited to reduce mass and cost, or to design larger and taller turbines that can generate more energy; in turn, all these effects may lead to a reduction of the cost of energy.

Traditional wind turbine controllers rely on feedback measurements to drive blade pitch, generator torque and yaw. Since they operate based on the response of the system as expressed by live measurements, these controllers are only capable of reacting to wind disturbances that have already impacted the wind turbine. This is an intrinsic limitation of all feedback-based mechanisms: since control actions are based on past measurements, the controller is always "late", in the sense that it reacts to events that are already taking place. To improve on this situation, control systems can be augmented with preview information, which informs the controller on the wind that will affect the turbine in the immediate future.

Wind preview can be obtained from turbine-mounted *light detection and ranging* (lidar) sensors, which are capable of measuring various properties of the incoming flow field up to several hundred meters in front of the rotor. Lidar-augmented control strategies are generically termed lidar-assisted control (LAC).

Several LAC formulations have already been investigated, and their performance in terms of power capture and load mitigation are reported in the literature. Bossanyi et al. (2014) describe a standard feedback controller enhanced by a feedforward blade pitch branch enabled by lidar wind preview. Results indicate promising reductions in blade flap and tower fore-aft fatigue

damage, without any appreciable loss in power production. Similar benefits are described by other sources such as, for example, Dunne et al. (2011, 2012). Benefits have also been confirmed in the field (Schlipf et al., 2013c), albeit to the present date only on a small research wind turbine. Feedforward torque control strategies have also been investigated; results indicate marginal increments in mean power capture at the expense of high power and torque variations (Bossanyi et al., 2014; Wang et al., 2013; Schlipf et al., 2013). More advanced formulations, such as nonlinear model-predictive controllers (Schlipf et al., 2013b) or flatness-based controllers (Schlipf et al., 2014), have also been enhanced with lidar wind preview information. Promising results were reported in terms of load reductions and power increase, at the expense of a much higher computational cost, which makes real-time execution more challenging to achieve and test in the field (Scholbrock et al., 2016).

Even though the potential of LAC is widely recognized, the system-level benefits that LAC may possibly bring to the levelized cost of energy (LCOE) are still not fully understood. In general two strategies have been suggested for reducing LCOE by LAC (Schlipf et al., 2018). The first is the *retrofit strategy*, which consists in using lidars to extend the lifetime of a wind turbine that has already been designed and installed. For example, Schlipf et al. (2018) reported the extension of the lifetime of a tower by 15 years. A second strategy is the *integrated approach*, in which LAC is considered as part of the system from its very inception. The idea in this second case is that, by considering LAC within the design process, its full potential can be realized by translating the benefits of load reductions directly into an improved turbine. Indeed, the adoption of a holistic system-level design approach was identified as an opportunity to assess the cost-benefit tradeoffs among turbine, lidar and control system by two IEA Wind Tasks: Task 32 on wind energy lidar systems, and Task 37 on systems engineering for wind energy (Simley et al., 2018, 2020) .

This work aims at taking a first step in this direction, providing an initial rough assessment of the potential benefits of considering LAC in the sizing of the two primary components of a wind turbine, namely the rotor and tower. The present work refines and expands the study described in Canet et al. (2020). In a nutshell, this study tries to give a preliminary general answer to the following main research questions:

  – To which extent can design-driving constraints be relaxed by LAC?

  – What is the best way of reaping the benefits brought by LAC in the design of rotor and tower?

  – To make LAC beneficial at the system-level, is it necessary to improve its performance or reduce its cost?

The present investigation intentionally does not commit to a specific lidar hardware or control formulation. In fact, the effects of LAC are considered here through a load-reduction model, defined according to the average performance of LAC systems reported in the literature. To understand trends, rather than focusing on a specific case, this baseline average literature-sourced model is expanded to cover an optimistic and a pessimistic scenario, thereby providing a range of possible behaviors. The study is performed on three wind turbines, which differ for wind class, size and power rating. These three reference machines are reasonable representations of current wind turbines available on the market. Clearly, the application of a literature-sourced range of load-reductions to three very different machines cannot give final and precise answers, which would require dedicated turbine and control-specific analyses conducted with coupled LAC-turbine simulations. However, the present approach offers

a way of obtaining an initial preliminary assessment of the potential benefits of adopting LAC, and it helps pinpoint the most
promising applications that should be further analysed.

The paper is organized as follows. Section 2 describes the approach and the models used in the study, while Sect. 3 analyzes the potential benefits of integrating LAC in the design of the tower and rotor of three different reference wind turbines. The study considers mass (and hence cost) reductions of these two components, but also investigates the design of towers that are taller or have a longer lifetime, including the effects of the purchase and maintenance costs of the onboard lidar system.
Section 4 closes the paper by reporting and discussing the main conclusions of the study.

## 2   Approach

Figure 1 presents a graphical depiction of the approach used in the present work. In a first phase, each turbine model is analyzed using a baseline non-LAC controller. This analysis highlights the benefits of reducing some design-driving quantity, and indicates by how much that quantity could be improved before another effect starts driving the design. Based on this
information, a second phase of the analysis initially considers each turbine equipped with a LAC controller, and then exploits the obtained load-reduction benefits to perform a structural redesign. Finally, the improved design is subjected to an economic analysis, whose goal is to establish tradeoffs between weight savings made possible by LAC and the additional expenses due to the purchase and O&M costs of the lidar. More details on these analysis processes are provided in the next sections.

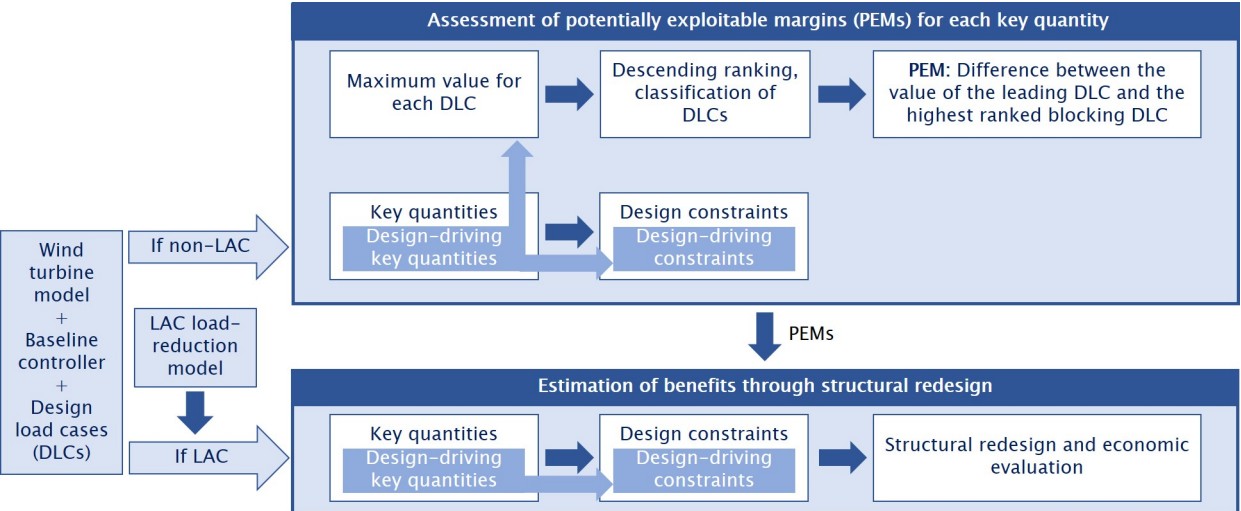

**Figure 1.** Approach overview.

## 2.1 Assessment of potentially exploitable margins

*Design-driving quantities* are those key indicators that define active constraints, thereby affecting the design solution. Design-driving quantities can be modified by LAC —or, more in general, by any control or technological solution— only to some extent, past which some other effect beyond the reach of LAC becomes the driver, preventing further improvements. The extent by which a design-driving quantity can be affected before another one becomes the driver is called here a *potentially exploitable margin* (PEM). It is an *exploitable margin* because, if it can be achieved, the design-driving constraints can be

relaxed and, therefore, the design can be improved. It is however only a *potential* margin because it represents an upper bound: in fact, a smaller improvement might be actually obtainable by LAC than this maximum limit.

A PEM is clearly a very valuable piece of information: there is no point in using LAC to reduce a certain quantity past the value where the driver switches to some condition that is not controllable by LAC. In fact, any further reduction would be futile, as it would not affect the design-driving constraints, and therefore the final design.

These considerations clearly do not apply exclusively to LAC, but more in general hold for any technology that has the potential to relax the design constraints of a system. Therefore, the analysis of PEMs is an extremely useful exercise, because:

- it is able to highlight the possible design benefits brought by the introduction of a new technology;

- it gives a target maximum margin of improvement that that technology should bring.

In the context of the current analysis, the assessment of PEMs is based on *key quantities* such as ultimate and fatigue loads,

and elastic deflections, which result from the aeroservoelastic simulation of a comprehensive set of design load cases (DLCs) run with a non-LAC controller. DLCs represent the different operating conditions that a wind turbine encounters throughout its lifetime, as defined by certification standards (IEC, 2005).

For the purposes of this work, DLCs are classified in two distinct groups: *modifiable* and *blocking*. In modifiable DLCs, the maximum value of each key quantity depends on the controller. For example, this is the case of loads obtained in power

production conditions (DLC 1.X). In fact, by modifying the pitch-torque controller of the turbine, the response of the machine changes, and consequently the loads that are produced also change. On the contrary, in blocking DLCs the key quantities are not affected by the controller. For instance, this is the case of loads generated in parked conditions (DLC 6.X). In fact, as the pitch-torque controller is not active when the turbine is parked, it clearly cannot influence the loads that are generated in that condition. Table 1 presents a classification of a selection of DLCs, including a description of the corresponding operating

condition.

PEMs are obtained via a two-step procedure.

First, the (active) design constraints that determine the sizing of a given wind turbine component are identified; these are termed *design drivers* or *design-driving constraints*. Design constraints are introduced in the structural design process of a wind turbine component to guarantee structural safety during its lifetime, ensuring that admissible values for stress, strain and

fatigue damage are never exceeded. Additional constraints are enforced to guarantee a safe clearance and to avoid collisions between blade and tower, to prevent buckling, and to ensure all other desired characteristics from the resulting design (Bot-

**Table 1.** Classification of a selection of the design load cases into *modifiable* and *blocking* (see text for a definition). NTM = Normal turbulence model; ETM = Extreme turbulence model; ECD = Extreme coherent gust with direction change; EWS = Extreme wind shear; EOG = Extreme operating gust; EWM = Extreme wind speed model.

| Classification | DLC | Design situation | Wind speed | Wind profile | Other condition |
|---|---|---|---|---|---|
| *Modifiable* | 1.1 | Power production | $V_{in}$:$V_{out}$ | NTM | |
| | 1.2 | Power production | $V_{in}$:$V_{out}$ | NTM | |
| | 1.3 | Power production | $V_{in}$:$V_{out}$ | ETM | |
| | 1.4 | Power production | $V_{rated} \pm 2\,\mathrm{ms}^{-1}$ | ECD | |
| | 1.5 | Power production | $V_{in}$:$V_{out}$ | EWS | |
| | 2.1 | Power production | $V_{in}$:$V_{out}$ | NTM | Grid loss |
| | 2.3 | Power production | $V_{out}, V_{rated} \pm 2\,\mathrm{ms}^{-1}$ | EOG | Grid loss |
| *Blocking* | 6.1 | Parked | $V_{ref}$ | EWM 50 year | Yaw mis. $\pm 8$ deg |
| | 6.2 | Parked | $V_{ref}$ | EWM 50 year | Grid loss |
| | 6.3 | Parked | $V_{ref}$ | EWM 1 year | Ext. yaw mis. $\pm 20$ deg |

tasso and Bortolotti, 2019). These constraints are functions of the key quantities resulting from the various DLCs, augmented by safety factors as prescribed by the norms. Other constraints, such as those enforced to avoid resonant conditions, are not dependent on DLCs.

Second, the maximum value of a key quantity is extracted from each considered DLC. The values are then sorted in descending order and labelled with the indication of the originating DLC. Each DLC is classified as modifiable or blocking. Clearly, the maximum value of a key quantity can only be reduced by LAC if its ranking is led by a modifiable DLC. The PEM is computed for each design-driving key quantity, and it is obtained as the difference between the quantity maximum value and the value of the highest ranked blocking DLC.

## 115    2.2    Estimation of benefits through structural redesign

PEMs can be exploited to improve the structural design of the wind turbine components that are driven by modifiable DLCs. To this end, DLCs should be run again, this time using LAC to yield new reduced values of the key quantities. However, as argued earlier on, instead of focusing on a particular case, it is more interesting to perform an analysis that is less specific and more general in character. To this end, a LAC load-reduction model was used here instead of re-running all DLCs with a given 120    LAC in the loop. The load-reduction model is simply represented by a set of multiplicative coefficients, which are defined for each key quantity associated with a modifiable DLC. Each coefficient expresses how LAC affects a key quantity with respect to a non-LAC controller; therefore, load reductions correspond to coefficients smaller than one in the model. Clearly, such coefficients depend on a multiplicity of factors, such as the specific control formulation, the tuning of its gains, or the performance of the lidar system. While a specific analysis is crucial when actually designing a wind turbine and its control 125    system, a specific analysis also clearly hinders somehow the generality of the results and conclusions that can be drawn from it. In this spirit, a range of possible performances —in contrast to a case-specific performance— is considered here by defining

different load-reduction scenarios. The load-reduction model and additional scenarios are based on results sourced from the literature, as more precisely discussed in §2.3.

The application of a LAC load-reduction model lowers some of the key quantities, in turn deactivating the associated design-driving constraints. To exploit the slack generated by LAC in the formerly active constraints, a redesign is performed to determine the structure that minimizes a desired figure of merit while guaranteeing structural integrity, in turn reactivating the constraints. After the redesign, an economic evaluation reveals the potential gains in LCOE, as discussed in §2.4.

## 2.3 LAC load-reduction model

The load-reduction model is based on a literature survey. The study reported in Bossanyi et al. (2014) was chosen as reference, because it presents a comprehensive list of the effects of LAC for several key quantities of various components. Additionally, that work was based on a rather standard controller, which might be representative of an initial conservative deployment on production machines. The implementation used a simple feedforward collective pitch LAC combined with a conventional feedback controller, applied to a 5 MW turbine. The paper reports a significant reduction of damage equivalent loads (DELs) resulting from DLC 1.2 for the blades, main bearing, tower top and tower bottom. Extreme loads resulting from extreme-operating gust conditions also experience significant benefits. On the other hand, power capture —and hence Annual Energy Production (AEP)— is largely unaffected by this LAC implementation.

The load-reduction model derived from Bossanyi et al. (2014) is reported in Table 2 for each component and modifiable DLC, in terms of percent changes with respect to a non-LAC controller. In the table, $F$ and $M$ respectively indicate force and moment components, expressed in the $(x, y, z)$ righthanded triad, where $x$ points downstream, $y$ is in the crossflow direction, and $z$ points vertically upwards. Components not reported in the table experience either null or negligible reductions.

The load-reduction model reported in Table 2 prompts a few important remarks.

First, the model only includes DLC 1.1, 1.2 and 1.3, which represent power production cases. In reality, these are not the only DLCs that are modifiable —in the sense that they can be affected by a change in the controller. In fact, additional modifiable DLCs are represented by DLC 1.4 (power production with extreme wind direction), 1.5 (power production with extreme wind shear), 2.1 (power production with control system fault or grid disconnection under normal turbulence conditions), and 2.3 (power production with control system fault or grid disconnection under extreme operating gusts). The first two of these DLCs are not considered in the LAC load-reduction model because they do not typically generate design driving loads, as further explained in §3.1. The case of DLC 2.1 and 2.3 is however different: here, maximum loads are typically generated during a shutdown, triggered by an extreme ambient condition change, a fault or a grid disconnection. When this happens, the entity of the generated loads will be largely dictated by the behavior of the shutdown procedure, which here is assumed not to be assisted by a lidar for safety reasons. On the other hand, loads generated during a shutdown might also depend to some extent on the state of the turbine at the time the shutdown was triggered, which does depend on the behavior of the LAC controller. A precise quantification of the effects of LAC on these DLCs would therefore require simulations with LAC in the loop, which are outside of the scope of the present preliminary work.

**Table 2.** Load-reduction coefficients based on Bossanyi et al. (2014), expressed as percentages with respect to a non-LAC controller.

| | BLADE | | | | | | |
|---|---|---|---|---|---|---|---|
| | Key quantity | Fx | Fy | Fz | Mx | My | Mz |
| DLC 1.2 | DEL | -3.8% | -0.1% | -0.25% | -0.4% | -3.8% | -3.5% |
| DLC 1.X | Extreme loads | | | | | -2.0% | |
| | Tip deflection | | | | | -2.0% | |

| | MAIN BEARING | | | | | | |
|---|---|---|---|---|---|---|---|
| | Key quantity | Fx | Fy | Fz | Mx | My | Mz |
| DLC 1.2 | DEL | -10.0% | | | -1.2% | -0.4% | -1.0% |
| DLC 1.X | Extreme loads | | | | | | |

| | TOWER TOP (YAW BEARING) | | | | | | |
|---|---|---|---|---|---|---|---|
| | Key quantity | Fx | Fy | Fz | Mx | My | Mz |
| DLC 1.2 | DEL | -12.0% | -0.1% | -2.1% | -2.0% | -1.8% | -0.2% |
| DLC 1.X | Extreme loads | | | | | | |

| | TOWER BOTTOM | | | | | | |
|---|---|---|---|---|---|---|---|
| | Key quantity | Fx | Fy | Fz | Mx | My | Mz |
| DLC 1.2 | DEL | -3.0% | 0.2% | -2.2% | -0.1% | -12.0% | -0.2% |
| DLC 1.X | Extreme loads | | | | | -5.0% | |

This point however leads to a second, more general, observation: the model in fact includes both DELs and extreme loads, neglecting lidar faults and assuming a lidar availability of 100%. While faults and availability (as long as it is not excessively low) will not impact DELs significantly, the situation is much more complicated for extreme loads. In fact, the malfunctioning of a lidar might in principle generate increases in ultimate loads, compared to a non-LAC case. A precise analysis of the possible faults and their consequences is clearly not only complex, but also highly case-specific. A mitigation of negative effects caused

by faults could be achieved, for example, through triple modular redundancy (Koren and Krishna, 2020), which would however clearly affect costs. A comprehensive analysis of these effects is outside of the scope of the present simplified study, and fault-induced increases of ultimate loads are therefore neglected here. Although this is an apparently strong assumption, in the end it does not affect the results of this study. In fact, as shown later, the benefits of the present LAC model on the turbines considered here are confined to fatigue mitigation, and hence only fatigue-driven components do benefit from LAC in this study. At a more

general level, one could wonder whether system-level benefits could be obtained by using LAC also for components driven by ultimate loads. While this remains an open question for now —as the present work is not able to provide definitive answers— it is clear that such an approach drastically raises the bar in terms of the complexity of the analysis and of the implementation, because of its obvious safety-related implications.

    Third, differences in the formulation and tuning of a LAC controller will generally imply different reductions of key quanti-

ties. To estimate these effects, the results obtained from various authors were compared. The most complete set of results was

found for DLC 1.2 in terms of DELs for fore-aft tower bending at tower bottom (FATBMTB), flapwise blade root moment (FBRM) and shaft torsional moment (STM), as reported by Schlipf et al. (2014); Bottasso et al. (2014); Haizmann et al. (2015); Schlipf et al. (2015); Schlipf (2016); Sinner et al. (2018). Table 3 reports the outcome of this analysis. There is a significant scatter in the results, especially for DEL FBRM and DEL STM, because of the variety of controller formulations and target wind turbine models. For instance, for DEL STM Schlipf et al. (2014) report a load reduction of 30% using a flatness-based feedforward controller, while Schlipf (2016) reports an improvement of 6% when using a feedforward-feedback controller. The lower values reported in Bossanyi et al. (2014) are most likely caused by the utilization of a fairly simple controller.

**Table 3.** LAC-enabled load reductions from Bossanyi et al. (2014) compared to other references.

|  | Bossanyi et al. (2014) | Additional literature |
| --- | --- | --- |
| DEL FABMTB | -12% | -16.4% $\pm$ 9.1% |
| DEL FBRM | -3.8% | -13.4% $\pm$ 6.6% |
| DEL STM | -1.2% | -11.8% $\pm$ 9.3% |

The scatter shown in Table 3 motivates the definition of two additional sets of coefficients that represent *optimistic* and *pessimistic* scenarios and provide a more general view of the benefits of LAC. The optimistic scenario is obtained by multiplying the baseline coefficients by a factor of 1.5, whereas the pessimistic one is obtained by using a factor of 0.5. Here again, it is worth remembering that the present study does not target one specific LAC controller, but aims at understanding basic trends.

A distinction must be made between the application of load-reduction coefficients to ultimate loads and deflections, which is straightforward (with the caveat of the effects of faults, as previously discussed), and to fatigue loads. The former simply consists in the correction of the key quantities obtained by a non-LAC controller with the corresponding coefficients of the load-reduction model. Combined loads —for example at tower base or at the main and blade pitch bearings— are computed from the corrected individual load components.

For fatigue damage, the following procedure is used. Site-weighted DELs are computed as

$$\text{DEL} = \sum_{v=V_{\text{in}}}^{v=V_{\text{out}}} f(v) L_{eq}(v), \tag{1}$$

where $f(v)$ is the Weibull probability density function at a wind speed $v$, while the damage equivalent load at that same wind speed is expressed as

$$L_{eq} = \left( \frac{\sum_{i=1}^{n} S_{r,i}^{m}}{N_{eq}} \right)^{1/m}, \tag{2}$$

where $m$ is the Wöhler coefficient, $S_{r,i}$ is the load range of a cycle $i$, $n$ is the total number of cycles and $N_{eq}$ the equivalent number of cycles (Hendriks and Bulder, 1995).

To compute LAC-reduced DELs, it is assumed that load reductions are independent of wind speed and load range. This way, the Weibull-weighted DEL reductions reported in the literature can be applied directly to the load time histories obtained here

with a non-LAC controller by aeroelastic simulations. Clearly this is an approximation, as LAC-enabled reductions generally depend on the wind speed, as reported by several studies (Bottasso et al., 2014; Schlipf et al., 2018, 2013). However, it was verified by aeroelastic analyses that this assumption does not significantly affect the results when the reduction coefficients are small, as those reported in Tables 2 and 3. For example, with reference to Table 3, considering the DEL FBRM reduction of -3.8%, the difference in fatigue margin at the blade root between wind speed dependent and independent reductions was found to be less than 2%; for the DEL FABMTB reduction of -12%, the fatigue margin difference at tower base was found to be approximatively equal to 5%. Given the character of this study, these differences were deemed to be acceptable and well within the margin of uncertainty of the analysis.

To complete the calculation of LAC-reduced DELs, transient combined loads are computed from the relevant components (for example, combining fore-aft and side-side components at tower base, and similarly combining the associated components at the main and pitch bearings), and then processed by rainflow counting to obtain DELs, finally searching for the point in the cross section of interest with the maximum damage. The computation of fatigue margin constraints for the steel tower is performed following the European regulations (Eurocode 3, 2005).

## 2.4 Economic evaluation

During the redesign phase, the components are evaluated from an economic point of view through suitable cost models, based on the characteristics of the wind turbine. The 2015 NREL cost model (WISDEM, 2020), which is an updated version of the 2006 model (Fingersh et al., 2006), is used for onshore machines, whereas the INNWIND cost model (Chaviaropoulos et al., 2014) is used for offshore turbines. The blade cost for both onshore and offshore models is computed based on the SANDIA cost model (Griffith and Johans, 2013). All cost model estimates are expressed in 2020 Euros (€), inflated by the consumer price index and exchange rate. The comparison of the various designs is based on LCOE, which is computed as

$$LCOE = \frac{FCR \cdot ICC}{AEP} + AOE, \tag{3}$$

where FCR [-] is the Fixed Change Rate, ICC [€] the Initial Capital Cost, AEP [MWh] the Annual Energy Production, and AOE [€/MWh] the Annual Operating Expenses.

## 2.5 Design and simulation environment

Aeroelastic analyses are performed with the Blade Element Momentum (BEM) based aeroelastic simulator `Cp-Lambda` (Bottasso et al., 2016), coupled with a conventional non-LAC controller (Riboldi et al., 2012). The aeroelastic simulator `Cp-Lambda` is also the core of the wind turbine design suite `Cp-Max` (Bottasso and Bortolotti, 2019; Bortolotti et al., 2016). This code can perform the combined preliminary optimization of a wind turbine, including both rotor and tower sizing.

The optimization of the blade aeroelastic characteristics can be divided into two coupled sub-loops, which size the external aerodynamic shape and the structural components. In this work, the aerodynamic shape of the blade is kept frozen, and the rotor is redesigned only from the structural point of view. The blade structural optimization algorithm aims at minimizing cost, while guaranteeing structural integrity and other requirements by enforcing a set of constraints that include, among others,

extreme conditions, fatigue damage, buckling, tower clearance, frequency placement, manufacturability and transportation. The optimization variables include the thickness of the structural elements (skin, spar caps, shear webs) for given blade layout and materials. The inertial and structural characteristics of each blade section are computed with the 2D finite element cross-sectional analysis code `ANBA` (Giavotto et al., 1983).

The structural sizing of the tower aims at minimizing its cost, while satisfying constraints from extreme loads, buckling, fatigue damage, as well as geometric constraints for manufacturing and transportation. The optimization variables include the diameter and thickness of the different tower segments for given material characteristics.

The formal description of the design algorithms can be found in Bottasso et al. (2012) and Bortolotti et al. (2016). Optimization is based on Sequential Quadratic Programming (SQP), where gradients are computed by means of forward finite differences.

## 3 Results

The potential benefits of adopting LAC in the early stages of the design of the rotor and tower of different wind turbines are analyzed next, following the approach described in Section 2.

### 3.1 Reference machines

Three reference wind turbines are considered: WT1, an offshore class 1A developed in Bottasso et al. (2016) as an evolution of the original DTU 10 MW reference wind turbine (Bak et al., 2013); WT2, an onshore class 2A (Bortolotti et al., 2016); and WT3, an onshore class 3A (Bortolotti et al., 2019). The principal characteristics of these machines are reported in Table 4, while additional details can be found in the corresponding references. These turbines are reasonable representatives of current products available on the market. The three machines have blades made of a glass-reinforced polymer and towers made of thin-walled tubular tapered steel sections.

**Table 4.** Principal characteristics of the three reference turbines.

| Turbine | WT1 | WT2 | WT3 |
|---|---|---|---|
| IEC Class & Category | 1A | 2A | 3A |
| Rated electrical power [MW] | 10 | 2.2 | 3.4 |
| Type | Offshore | Onshore | Onshore |
| Rotor diameter [m] | 178.3 | 92.4 | 130.0 |
| Specific power [W/m$^2$] | 400.5 | 298.3 | 252.4 |
| Hub height [m] | 119.0 | 80.0 | 110.0 |
| Blade mass [t] | 42.5 | 8.6 | 16.4 |
| Tower mass [t] | 628 | 125 | 553 |

Table 5 compares the three machines in terms of capital cost (CAPEX), operational expenses (OPEX), AEP, and LCOE with some actual installations in the United States according to Stehly et al. (2017). The cost breakdown is expressed in 2017

United States Dollars (USD), and CAPEX does not include financial costs. The comparison shows a good match between the costs of the onshore 2.2 MW WT2 turbine and the 2017 US land-based 2.32 MW machine. The costs of the 3.4 MW WT3 turbine, even if slightly higher for some figures, are also in reasonable agreement with the US reference. For the offshore case, a bottom-fixed 5 MW machine is compared to the 10 MW used in the present study. Larger differences are found here, for instance in the OPEX costs, due to the very different rating of the two turbines, although the LCOEs are relatively similar.

**Table 5.** Cost breakdown of the different reference models expressed in 2017 USD.

| Cost [USD/kW] | Onshore | | | Offshore | |
|---|---|---|---|---|---|
| | Stehly et al. (2017) | WT2 | WT3 | Stehly et al. (2017) | WT1 |
| Rating [MW] | 2.32 | 2.2 | 3.4 | 5 | 10 |
| CAPEX [USD/kW] | 1454 | 1297 | 1759 | 3846 | 4379 |
| OPEX [USD/kW] | 43.6 | 48.1 | 51.4 | 144 | 225 |
| AEP [MWh/MW] | 3633 | 3520 | 3866 | 3741 | 4500 |
| FCR [%] | 7.9 | 7.9 | 7.9 | 7.0 | 7.0 |
| LCOE [USD/MWh] | 43.6 | 42.9 | 49.2 | 110.5 | 118.1 |

## 3.2    Assessment of potentially exploitable design margins

The present study considers a reduced set of DLCs (IEC, 2005), which are responsible for generating the design drivers of these machines (Bottasso et al., 2016; Bortolotti et al., 2016, 2019). The set includes power production with normal turbulence (DLC 1.1 and DLC 1.2), extreme turbulence (DLC 1.3), loss of electrical network in normal turbulence (DLC 2.1) and with extreme operating gusts (DLC 2.3). Additionally, parked conditions are also considered in yaw misalignment (DLC 6.1), with

grid loss (DLC 6.2) and with extreme yaw misalignment (DLC 6.3).

### 3.2.1    Tower

A first analysis of the design-driving key quantities and constraints of the three towers unveils a significant potential that could be exploited by LAC.

For the design constraint analysis, several cross-sections are considered along the tower height, where three local conditions

are evaluated: buckling, ultimate strength based on von Mises stresses, and fatigue damage. Additionally, the placement of the first fore-aft and side-side frequencies is constrained to avoid crossing the one-per-rev at rated rotor speed.

For simplicity of discussion, only results at the tower top and bottom cross-sections are shown in Fig. 2, where the constraint margins are displayed. These are formulated as the relative difference between the local conditions and their admissible values. A null value therefore indicates an active constraint, whereas a positive value indicates a slack condition, i.e. a constraint that

is satisfied but inactive.

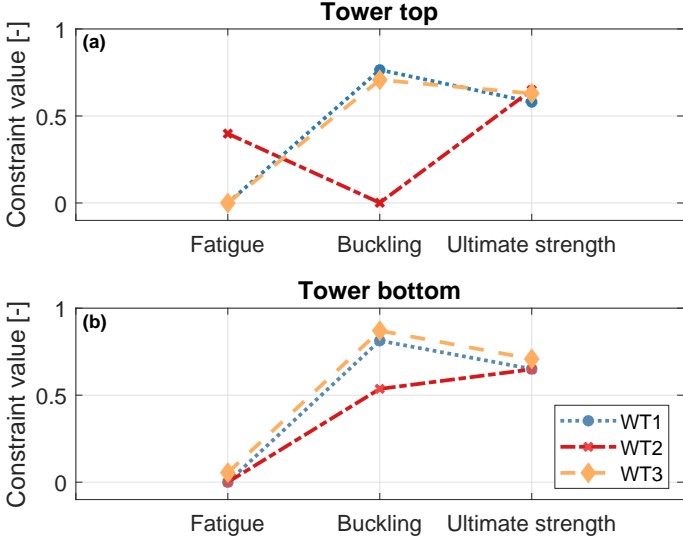

**Figure 2.** Design constraints at tower top **(a)**, and tower bottom **(b)**.

Considering first the tower top section, Fig. 2a shows that at this location the towers of WT1 and WT3 are driven by fatigue, whereas buckling and strength are well below their maximum allowed values. The design of this section can therefore benefit from reductions in fatigue damage, which is mostly produced by the modifiable DLC 1.2 (power production in normal turbulence). On the other hand, the upper section of the WT2 tower is driven by buckling, whereas fatigue damage and ultimate
strength are inactive. The PEM at this position along the tower is related to the combined bending moment (CBMTT). The rankings of this key quantity for the three turbines are shown in Fig. 3a. All values are normalized with respect to the leader and, for clarity, only the leading and first blocking DLCs are shown. The ranking for WT2 is led by DLC 1.3, a modifiable DLC. The first blocking DLC is 2.1, which appears at position 28 in the ranking, leading to a PEM of about 20%.

Considering the tower bottom cross-section, Fig. 2b indicates that all three towers are driven by fatigue. Load rankings
for combined bending moment at tower bottom (CBMTB) are reported in Fig. 3b. Results show no potential reduction for the extreme-load constraints, since the load rankings of the WT1 and WT2 towers are led by blocking DLCs. A PEM of about 21% is visible for the WT3 tower, which however cannot be exploited since extreme loads do not drive the design at this section.

### 3.2.2 Rotor

Rotor design constraints include limits on the placement of the lowest natural frequencies to avoid resonant conditions, and
290 a safe clearance with respect to the tower. Additionally, several cross-sections are considered along the length of the blade, where upper limits for strains, stresses and fatigue damage are prescribed on the spar caps, shell and shear webs. An excerpt from this extensive set of constraints is shown in Fig. 4; the shell, spar cap and shear web constraints are shown only at the midspan section of the blade, for simplicity of illustration.

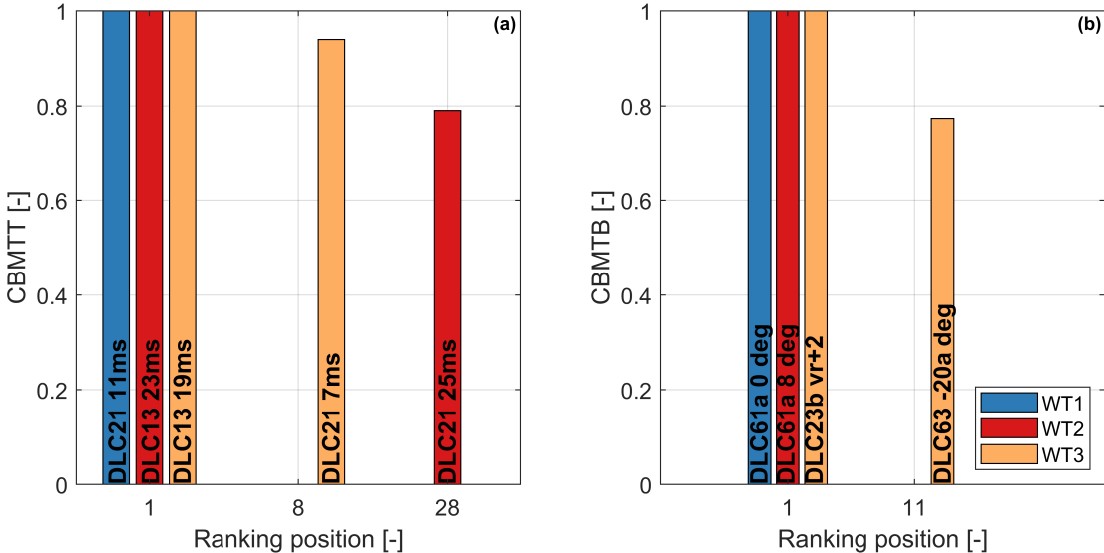

**Figure 3.** Ranking of normalized combined bending moment at tower top (CBMTT) **(a)** and tower bottom (CBMTB) **(b)**, for the three turbines. Only the leading and first blocking DLCs are shown.

The spar caps are the components that play the largest role in dictating the overall blade mass, as they mainly provide the blade flapwise bending stiffness. The design of these elements is driven by the blade-tower clearance constraint, which limits the maximum blade tip displacement (Fig. 4a). On the other hand, stress, strain and fatigue constraints are all inactive (Fig. 4b). The tip displacement rankings, shown in Fig. 5a, indicate a significant reduction potential for all turbines, since they are all led by modifiable DLCs. This key quantity for all three turbines is first blocked by DLC 2.1, leading to PEMs between 8% (WT1, ranking position 7) and 21% (WT2, ranking position 28).

The sizing of the shell is mainly driven by the fatigue damage constraint (Fig. 4c). This is also the main driver in the design of the shear webs, which are elements made of sandwich panels that carry shear. Fatigue damage is driven by the modifiable DLC 1.2. However, here the reduction potential is limited by technological constraints that bound from below the thickness of these elements. The load ranking of the combined blade root moment (CBRM) is shown in Fig. 5b, highlighting potential reductions. Indeed, all turbines are again first blocked by DLC 2.1, with large PEMs for WT2 (25%, ranking position 2) and WT3 (30%, ranking position 3).

### 3.3 Estimated benefits through structural redesign with LAC

This section aims at quantifying the benefits of integrating LAC within the design of the blade and tower of the three reference wind turbines. To this end, the rotor and tower of each turbine are reoptimized, considering loads and elastic deflections as reduced by the coefficients of the load-reduction model (Table 2) and the additional optimistic (values incremented by 50%) and pessimistic (values reduced by 50%) scenarios. The economic evaluation is performed as indicated in Section 2.4, considering

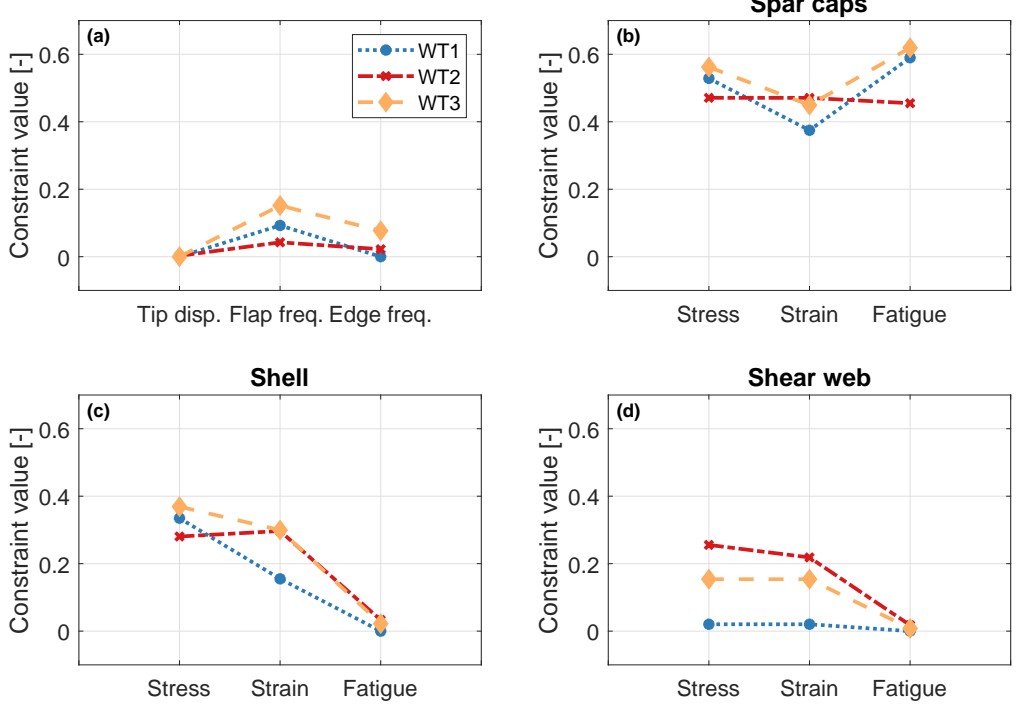

**Figure 4.** Rotor design constraints **(a)**. For a midspan section of the blade, design constraints at the spar caps **(b)**, shell **(c)**, and shear webs **(d)**.

a fixed change rate (FCR) of 7%. It is further assumed that two lidar scanners have to be purchased over a turbine lifetime of 20 years. This results in an additional 100,000 € of ICC. Furthermore, the AOE includes an additional 2,500 €/year of lidar O&M cost. These costs have been estimated based on input from two major lidar manufacturers, and only include hardware-related costs. Due to a lack of information, the costs of development or licensing of LAC control software, related commissioning and software maintenance have been neglected.

### 3.3.1 Tower redesign

Figure 6 reports changes in the LAC-based redesigned towers with respect to the initial baselines, when the tower height is held fixed. The solid color bars correspond to the nominal load-reduction model, while whiskers indicate the effects of considering the pessimistic and optimistic scenarios. To ensure direct comparability with the baselines, the redesigned towers are considered to be made of several thin-walled tubular tapered steel sections. Additional geometric constraints to ensure realistic tower shapes are also considered.

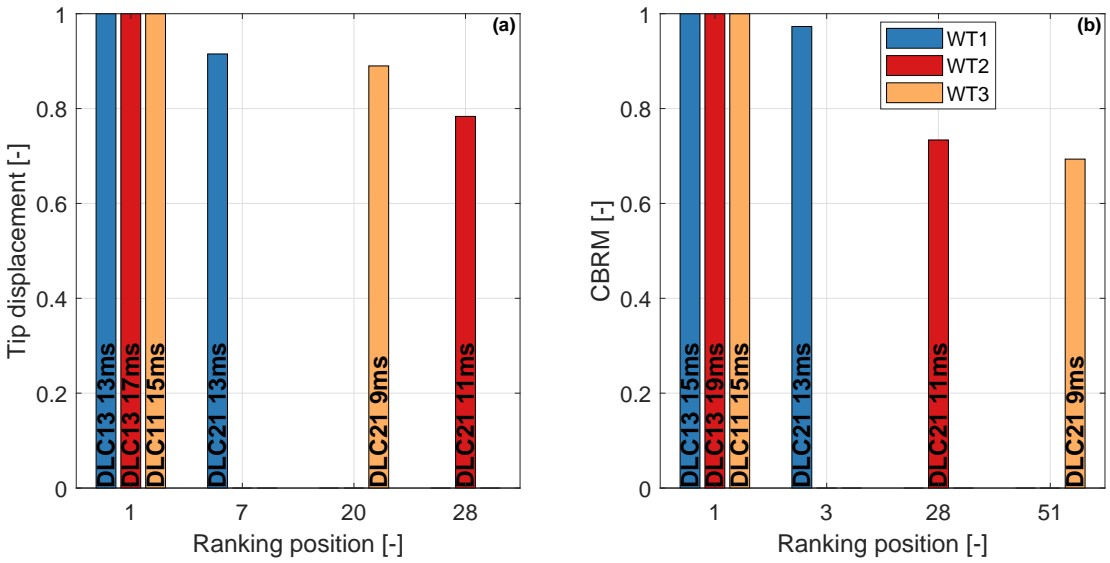

**Figure 5.** Ranking of normalized blade tip displacement **(a)**, and combined blade root moment (CBRM) **(b)**, for the three turbines.

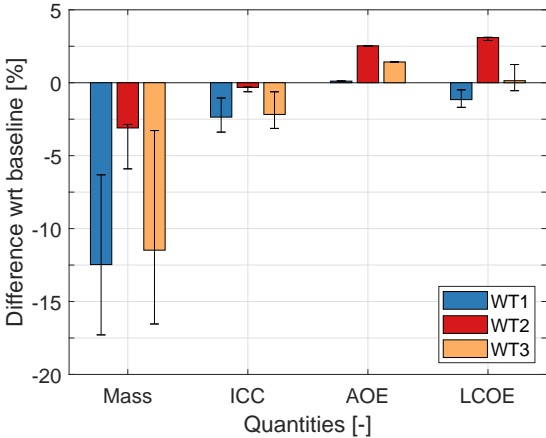

**Figure 6.** Effects of LAC on the redesign of the tower with respect to the initial baselines. Solid bars: load-reduction model of Table 2; whiskers: range of the pessimistic and optimistic scenarios.

Both towers of WT1 and WT3 enjoy significant benefits from large reductions in fatigue damage, which decrease mass between 5% for the pessimistic scenario and 17% for the optimistic one. In turn, the lighter weight induces significant reductions in the ICC of both turbines. On the other hand, annual operating expenses (AOE) show a different behavior. Indeed, the additional expenses generated by the maintenance of a lidar system do not significantly add to the already high O&M costs of the offshore turbine WT1. For the onshore machines WT2 and WT3, where these costs play a larger role, AOE increases by approximately 2%. For all turbines, AEP is essentially unaffected. In the end, the combination of these various effects produces a reduction in LCOE of about 1.2% for WT1, and a very slight increase of 0.1% for this same figure of merit for WT3 (Fig. 6).

The WT2 tower presents a different trend. Indeed, the upper segment of this tower is driven by buckling and CBMTT presents a significant PEM of about 20% (see Fig. 3a). However, this PEM can not be exploited, since the LAC load-reduction model (Table 2) does not affect extreme loads at tower top. As a consequence, the redesign is only capable of a limited mass reduction that, in combination with the significant lidar costs, leads to an increase in LCOE.

### 3.3.2 Taller tower redesign

Instead of reducing tower mass (and hence cost), LAC-enabled improvements in fatigue damage and ultimate loads can be exploited to design taller towers. In fact, by reaching higher above the ground, the rotor is exposed to faster wind speeds, thus increasing AEP; thanks to LAC, this can be achieved without significantly increasing the cost of the tower. To explore the effects of this concept, towers of increasing heights were designed. The study assumes that LAC performance does not depend on tower height. To ensure direct comparability, the redesigned towers are also considered to be made of several thin-walled tubular tapered steel sections. The corresponding geometrical constraints are therefore also included in the redesign problem.

The study is here performed in two steps. First, the tower structure is sized with a non-LAC controller for a given height. The design objective is minimum mass, constrained to guarantee structural integrity. Next, the resulting tower design is reoptimized considering the different scenarios of the LAC load-reduction model, exploiting the slack that it generates in some design-driving constraints. The procedure is repeated for increasing tower heights, until no further improvements are possible, or an upper limit of 15% height increase with respect to the baseline is reached.

The effects on mass, ICC, AEP, AOE and LCOE for the three reference machines are reported in Fig. 7.

Different trends are observed for the three turbines. The tower of the offshore machine shows a large potential: for each of the analyzed heights, mass reductions with respect to the non-LAC configuration always translate into decreases in ICC. At the same time AEP increases, whereas AOE remains mostly constant due to the already high O&M costs. LCOE decreases gradually as tower height is increased. However, most of the gains are already achieved for a height increase of 5%, which is associated with an LCOE decrease of about 1.5% (Fig. 7e).

An opposite trend is obtained with the tower of WT2: because of its different design drivers, this machine does not benefit from a taller tower, as already noted in §3.2.1. However, the trend indicates that some LCOE improvements might be possible for very tall towers, which were however deemed unrealistic past the upper bound of a 15% height increase.

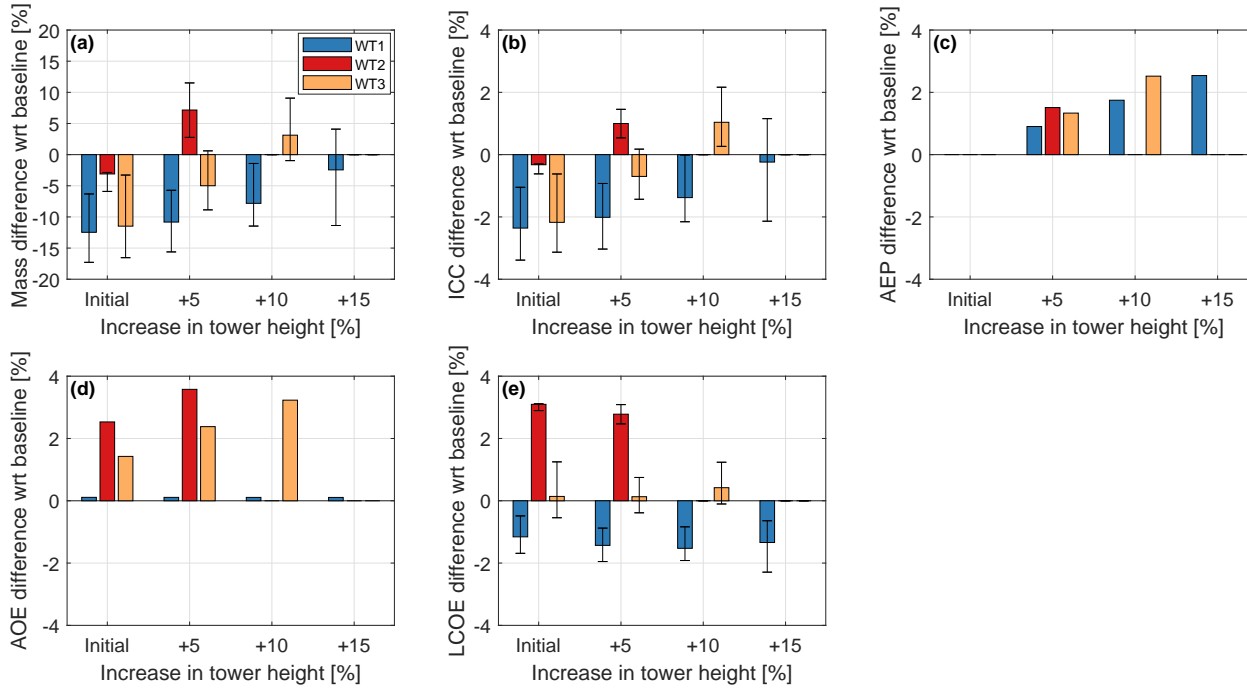

**Figure 7.** Effects of LAC on the redesign of towers of increasing height with respect to the initial non-LAC baselines. Solid bars: load-reduction model of Table 2; whiskers: range of the pessimistic and optimistic scenarios. The study considers increments of +5%, 10% and 15% in tower height for WT1; an increment of 5% in tower height for WT2; and increments of 5% and 10% in tower height for WT3.

Similarly, a taller tower appears not to be very promising even for the onshore fatigue-driven WT3 turbine, although for different reasons. Here, although a 5% height increase lowers tower mass and ICC and improves AEP by about 2%, these benefits are offset by an increase in AOE, resulting in marginal —if not completely negligible— benefits in LCOE.

### 3.3.3 Tower redesign for longer lifetime

Instead of aiming for less expensive or taller towers, as done so far, yet another way to try and exploit the load benefits brought by LAC is to extend the tower lifetime. In this case, the baseline towers are first designed for a 20-year lifetime based on the key quantities resulting from a non-LAC controller. Here again, the towers are redesigned for increasing lifetime in two steps. First, the tower structure is sized with a non-LAC controller for a given lifetime. Next, the resulting tower is reoptimized based on key quantities modified by the LAC load-reduction model (Table 2). WT2 is excluded from this analysis, because of the very limited relevance of fatigue in the sizing of its tower, as shown earlier. To ensure direct comparability with the baseline, the redesigned towers are considered to be made of several thin-walled tubular tapered steel sections, and the corresponding geometrical constraints are included in the sizing.

The tower mass of both WT1 and WT3 increases substantially when sizing for a longer lifetime without using LAC. This negative effect is very nicely counteracted by the use of LAC. Figure 8 reports mass changes generated by LAC for increasing lifetime; all results are computed with respect to initial non-LAC 20-year baselines. At a lifetime of 40 years, which is double the conventional life duration, the tower mass of WT1 is still 10% lower than for the non-LAC 20-year case. The effect is similar, although a bit less pronounced, even for WT3: for a lifetime of 40 years with LAC, this tower has in fact nearly the same mass of the 20-year non-LAC design.

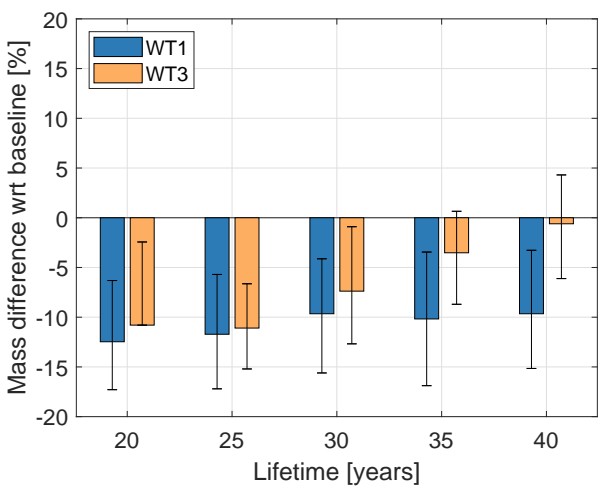

**Figure 8.** Effects of LAC on the redesign of towers of increasing lifetime with respect to 20-year non-LAC baselines. Solid bars: load-reduction model of Table 2; whiskers: range of the pessimistic and optimistic scenarios.

It should be remarked that these trends are obtained under the assumption of a 100% lidar availability; additionally, because of the approximations implicit in the assumed load-reduction model, these results can only be regarded as preliminary rough trends. However, the use of LAC to design towers with longer lifetimes seems to be much more promising than the alternative strategies of aiming for reduced costs or improved AEP by taller towers. Indeed, the trends shown here are in line with the results reported in Schlipf et al. (2018), which estimated a 15-year extended lifetime for a tower without redesign. Additionally, since the tower cost plays a large role in ICC, reductions in LCOE could be expected by the installation of towers with a longer lifetime. Alternatively, the towers could be reused to support more modern rotor-nacelle assemblies, playing the role of long-term support structures that do not necessarily have to be upgraded at the same pace of the rest of the turbine.

### 3.3.4 Rotor redesign

Only rather modest mass reductions are achieved for the blades of all models and for all scenarios, due to the moderate influence of LAC in design-driving constraints. The situation is more precisely illustrated by Fig. 9, which shows the largest improvements for WT1 and essentially no effect for WT2.

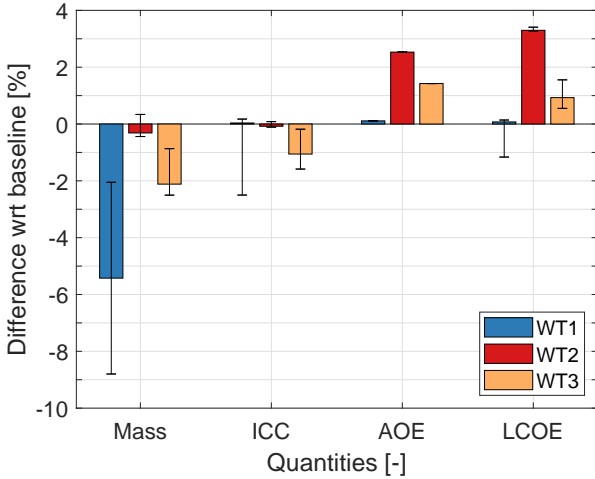

**Figure 9.** Effects of LAC on the redesign of the rotor with respect to the initial baselines. Solid bars: load-reduction model of Table 2; whiskers: range of the pessimistic and optimistic scenarios.

Indeed, the LAC load-reduction model reported in Table 2 shows a larger effect of LAC in fatigue damage mitigation than
in the reduction of ultimate loads and deflections. Although shell and shear webs are both driven by fatigue, they are already thin structures with limited reduction potential before technological constraints on their thickness become active. In turn, this leads to the fatigue PEMs not being fully exploited. The design of the spar caps is also not significantly affected by LAC. In principle, a significant PEM is present for tip deflection, but unfortunately here again the LAC load-reduction model has only modest 2% improvements for this key quantity. Addionally, as previously noted, the exploitation of the reduction of an ultimate
condition by LAC raises important issues related to safety, and might imply drastically increased costs to ensure redundancy.

For all three turbines, the reduction in ICC generated by the use of LAC in the redesigned rotors is not significant enough to compensate for the increase in AOE. Therefore, LCOE increases for all onshore machines and decreases in a negligible way for the offshore turbine.

## 3.4 Cost sensitivity analysis

Finally, a sensitivity analysis is performed to understand to what extent the purchase and maintenance costs of a lidar system can influence the reduction in LCOE. Baseline values of 100,000 € and 2,500 €/year, respectively for purchase and maintenance, are gradually modified until reaching the limit of $\pm 100\%$ variations. It is assumed that lidar-related yearly maintenance costs are constant throughout the wind turbine lifetime, and are therefore not affected by external factors, such as the replacement of the lidar system. Purchase price includes both the cost and the number of lidar systems required throughout the wind turbine
lifetime. The analysis considers the nominal LAC load-reduction model of Table 2 applied only to WT1 and WT3, as WT2 did not seem to have any real potential for improvement. Clearly, redundancy to ensure safety would significantly increase all of these costs.

It should be noticed that purchase and maintenance costs are treated here as two independent variables. In reality, purchase price could be correlated with performance, and therefore it might affect load reductions. Additionally, purchase price could be correlated with maintenance: a higher cost of the lidar could imply a more sophisticated device, which might be more costly to maintain, but it could also be correlated with build quality, which then might be inversely related to maintenance cost. Such considerations would require a sophisticated cost model of the lidar, which was however unfortunately not available for this research. The present analysis, being based on the simple change of the two independent quantities purchase and maintenance costs, could then be interpreted as a price positioning study, where the lidar manufacturer tries to understand the correct price range for the device to make it appealing to customers.

Figure 10a shows that only a modest effect in LCOE can be achieved for WT1 when purchase and maintenance costs are modified. On the other hand, an order of magnitude larger effect is observed for WT3 (Fig. 10b), where the incidence of the lidar-associated costs is more prominent given the smaller size and rating of this turbine.

Break-even is indicated in both figures as a dotted line, located in the white area that separates reductions (blue) from increments (red) in LCOE. The break-even line is almost perpendicular to the purchase cost axis, implying a large sensitivity of LCOE to this quantity. The figure shows that reductions in purchase costs appear more effective than reductions in O&M costs. This seems to indicate that lidar manufacturers should try to keep the cost of the device as low as possible. The fact that maintenance costs are less relevant might indicate that simple and cheap lidars —although possibly a bit more expensive to maintain— would be more appealing than sophisticated but expensive ones. Cheap single units, as long as availibility remains sufficiently high, might also be very interesting from the point of view of redundancy, which might open up the possibility of exploiting ultimate load reductions. However, as noticed earlier, more sophisticated models —capable of capturing the couplings among purchase price, performance (including availability), lifetime and maintenance— would be necessary to identify economically optimal development strategies for lidar systems.

Overall, results indicate that only modest reductions in LCOE are possible, even with very low LAC-based costs.

## 4   Conclusions

This paper has presented a preliminary general analysis on the potential benefits of integrating LAC within the design of the rotor and tower of a wind turbine. The design was performed as a constrained optimization based on aeroelastic simulations, conducted in close accordance with international design standards.

The benefits generated by the use of a lidar for controlling a turbine were quantified through a load-reduction model sourced from the literature, considering an average performance of the lidar-assisted controller and additional pessimistic and optimistic scenarios. This approach, in contrast to the use of an actual lidar-assisted controller in the loop, was chosen in order to draw conclusions on general trends, rather than on the effects of a specific LAC implementation. Realizing that any such redesign exercise is typically highly problem-specific, the study was conducted considering three representative turbines of different class, size and rating.

Based on the results of this study, the following conclusions can be drawn.

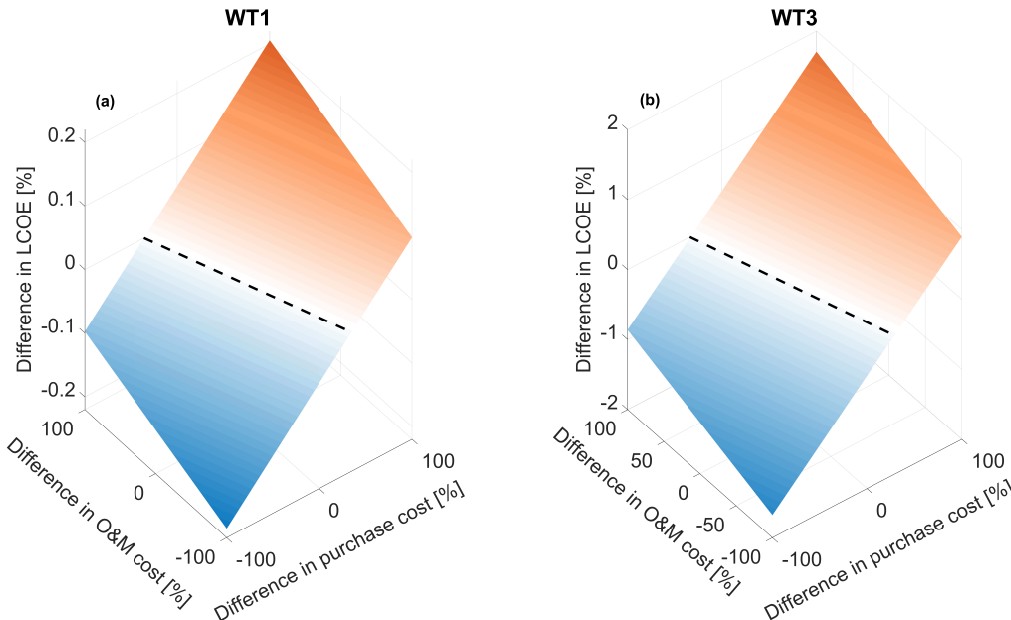

**Figure 10.** Percent variation of LCOE as a function of purchase and O&M costs of LAC systems for the offshore machine WT1 **(a)**, and the onshore machine WT3 **(b)**.

First, a significant improvement potential was observed when the design is driven by fatigue. Indeed, fatigue damage is primarily generated in power production in turbulent wind conditions. Here, the lidar-generated preview of the wind that will shortly affect the rotor is clearly beneficial: as the controller "sees" what will happen, it can anticipate its action. This is in contrast to the case of a pure feedback controller that, since it can only operate in response to a phenomenon that has already taken place, is by definition "late" in its reaction. In turn, the lidar preview information leads to a general reduction of load fluctuations, and hence of fatigue damage.

On the contrary, the improvement potential is only very limited for components driven by ultimate conditions (such as maximum stresses, strains or blade tip deflection). Indeed, these ultimate conditions cannot always be modified by LAC. In addition, even when LAC plays a role, other factors may have an even larger effect; for example, this is the case of shutdowns, where the pitch-to-feather policy may have a dominant role in dictating the peak response. But even when LAC does improve design-driving ultimate conditions, an even more general question still remains: shall one design a component based on an ultimate condition that was reduced by LAC? If so, what are the extra precautions that should be taken in order to hedge against faults, inaccuracies, misses, or unavailability of the lidar? These issues were not considered here, which is a limitation of the present study. However, it is possible that —at least in some of the cases analyzed in this work— the improvements to ultimate conditions brought by LAC would have to be completely discarded when these additional aspects are considered, or that extra costs would have to be added, for example to ensure redundancy by the use of multiple lidars.

It was also found that, for fatigue-driven towers, significant benefits in mass (on average equal to about 12%, for the cases considered here) can be obtained by the use of a LAC controller. However, these benefits are largely diluted by looking at the more general metric LCOE. In fact, only a large offshore machine showed improvements for this figure of merit: since O&M costs are already high for an offshore turbine, the extra costs due to the lidar play a lesser role. For smaller turbines the situation is different, and the benefits in mass do not repay for the costs of the lidar.

Instead of simply reducing mass, LAC can be used to either increase hub height (which increases power capture in sheared inflow) or to extend lifetime. Both approaches were considered here. The most interesting results were again obtained for fatigue-driven offshore towers. Indeed, a 15% taller tower was found to present approximately the same mass of the baseline, but with a 2% higher AEP. Even more interestingly, a LAC-enabled tower was designed with double the lifetime and 10% less mass than the baseline.

The situation for the rotor is less promising. In principle, spar caps —which are the main contributors to blade mass— could greatly benefit from LAC when tip deflection is the main driver. Here again lidar preview can clearly help when maximum deflections are triggered by strong wind gusts. On the other hand, stiffness requirements caused by the placement of the flap frequency can substantially reduce this margin of improvement, as this is a blocking effect. Additionally, one would have again to guarantee that the safety-critical tip clearance constraint is always satisfied during operation, which might require redundancy of the lidar or other safety measures. Shear webs and shell are often driven by fatigue, a condition that could in principle be exploited by LAC. However, the improvement potential is limited due to the already limited thickness of these components. In summary, the integration of LAC into the design of the rotor does not seem to lead to significant benefits in terms of LCOE.

Finally, a simple parametric study on the purchase and O&M costs of a lidar system was performed. As previously observed, the study shows that LCOE is largely independent from the LAC purchase and O&M costs in the offshore case. Although a larger effect is visible in the onshore case, improvements in LCOE caused by reductions in the lidar costs are still quite modest. This might indicate that, instead of targeting price reductions, lidar research and development should focus on performance. On the other hand, significant price reductions might allow for redundancy, which in turn would enable the targeting of drivers based on ultimate conditions.

The present work is based on a number of assumptions, and further work should be performed before more definitive conclusions can be drawn. First, only three turbines were considered; although these machines are reasonable approximations of contemporary products, it is clear that design drivers are typically turbine specific, and a more ample range of cases should be investigated. Additionally, only the conventional configuration of thin-walled steel towers with circular tubular tapered sections was considered. This configuration presents important geometric constraints that impact the benefits of LAC. Second, there was no attempt here to consider lidar availability, faults and possible redundancy; an analysis of these aspects would help in clarifying whether LAC-enabled reductions in ultimate conditions can indeed be exploited in the structural redesign of the blade and the tower or not. Finally, it should be remarked that the use of a generic load model implies some significant approximations. Although this was done here on purpose with the goal of making the study more general, it is also clear that the performance of different LAC systems can be very different, depending on the lidar characteristics and on the controller

formulation and tuning. Therefore, here again, more specific studies based on fully-coupled simulations should be performed to further explore the trends reported here and find additional niches of applicability of LAC missed by the present general analysis.

Notwithstanding the limitations of this study, in the end it appears that the answer to the question of whether LAC is beneficial or not might not be so clear cut, and in reality the situation is much more complex and varied (and also interesting). In hindsight, this is also a useful reminder that apparently obvious improvements do not always necessarily translate into real system-level benefits. For example, reducing some loads might be irrelevant if the design is driven by other factors, or might not pay off if the cost of that reduction neutralizes its benefits. This also stresses once more the central importance of systems
engineering and design for the understanding of the true potential of a technology.

*Author contributions.*  HC performed the analysis on potentially exploitable margins and conducted the design studies; SL prepared the lidar load-reduction model and assisted in the application of the model in the design framework; CLB formulated the analysis methodology based on the new concept of potentially exploitable margins, proposed the use of a generic lidar load-reduction model, and supervised the work; HC and CLB wrote the paper. All authors provided important input to this research work through discussions, feedback and by improving
the manuscript.

*Competing interests.*  The authors declare that they have no conflicts of interest.

*Acknowledgements.*  The authors acknowledge the participants of the IEA Task 32+37 workshop "Optimizing Wind Turbines with Lidar-Assisted Control using Systems Engineering" for the valuable discussions.

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
