# Peer review of "What are the benefits of lidar-assisted control in the design of a wind turbine?"

_Wind Energy Science, 2020_

## Referee Comment (RC1) · Anonymous Referee #1 · 19 Jan 2021

Interesting approach to a relevant question. Explanations sometimes not so easy to follow. Difficult to assess whether the conclusions can really be considered to be general. Dealing with design load cases, especially ultimate loads, can be very case-specific. Perhaps more emphasis should be given to the results giving useful indications, but that the issues should be looked at in detail for any specific case. (Line 109-110: yes, this approach makes the results less specific, but also less relevant to any particular case.)

Abstract wording, line 7: "with potentially benefits" suggest "with potential benefits", and in "essentially limited to the sole tower", suggest deleting "sole".

Load cases, Table 1: What about other fault cases? Further down it seems they are assumed to be unaffected, but often they need vary careful thought, and some may

prove to be troublesome. As an example, what about the case of ultimate loads when the lidar fails to provide a good signal for any reason, and the failure is not detected? There will always be times when the lidar signal is unavailable, when the controller should revert to a safe mode. I don't think this has been considered. If the failure is undetected, so safe mode is not activated, it is possible that the control will result in higher ultimate loads than without any lidar at all. This is just one example. The results should include a strong caveat about load cases that have not been considered but that may affect the conclusions. (Line 139: "Hence, LAC-induced load reductions were assumed to be null for these DLCs, which is a conservative choice" - it is not necessarily conservative - LiDAR could make things worse in some situations.)

It might be helpful to list the DLCs that are *not* considered, with reasons why not considered important (some will be obvious of course).

Line 115: "guaranteeing" (spelling)

Line 169: "it is assumed that load reductions are independent of wind speed" - seems a strange statement. Load reductions may be much greater above rated when pitch control is active, and can depend strongly on wind speed.

Figure 6: Make figures bigger to help legibility. Why are some bars missing?

Figure 7: It would be nice to display the non-LAC mass increases compared to 20 yr life, and how they change with LAC.

Line 429: "radar" should be "lidar"?

---

## Referee Comment (RC2) · Domenico Di Domenico (Referee) · 22 Jan 2021

The paper deals with the benefits of including LAC in turbine design, evaluated in terms of LCOE. It is shown that LAC can reduce the mass of the tower, increase its height (which increases production) or extend its operating life. The LAC also allows to reduce the stress on the blades and thus to modify the structural properties of the rotor. The study is conducted considering the "modifiable" DELs, i.e. the DEL for which the LAC has an impact on the results. To evaluate the benefits of the LAC, the DELs and extreme loads are scaled-up, based on the results of a study Bossanyi, presented in 2014. A strong simplifying assumption is made regarding DEL scaling, which assumes that the reduction is independent of the wind speed and load range. This work seems to suggest that the gain obtained by the design of the turbines including the LAC is

negligible in most cases, given the current costs of Lidar and its maintenance costs. The authors are aware of the limitations of the study, and it is explicitly stated that these results are strongly impacted by the its assumptions. Anyway, this issue (identified by the IEA Task 32) has a real interest, and this paper provides a good starting point for its understanding. In my opinion, an important improvement to this work would be to consider the variability of the gains given by the LAC (or even to determine from which gains LAC becomes economically interesting), as the potential benefits depend on the used control technique, on its settings, on the operating conditions and, obviously, on the NON-LAC technique to which LAC is compared. This would have the merit of identify a level of performance that LAC would have to reach in order to be economically interesting, and could strongly incentive further research efforts on this topic.

Minor remark: At some points the paper is a bit hard to read. In this sense, a schematic summary or table, and an effort to make the explanations simpler and clearer can make the paper more readable.

---

## Author Comment (AC1) · 5 Jul 2021

**REVISION TO MANUSCRIPT DRAFT**

**Wind Energy Science Discussion**

**What are the benefits of lidar-assisted control in the design of a wind turbine?**

The authors would like to thank the two reviewers for their time and for the useful feedback. All inputs that they provided have contributed to the improvement of the paper.

A list of point-by-point replies to the reviewers' comments is reported in the following.

**Reviewer #1**

1. **[Reviewer]** *Interesting approach to a relevant question. Explanations sometimes not so easy to follow. Difficult to assess whether the conclusions can really be considered to be general. Dealing with design load cases, especially ultimate loads, can be very case-specific. Perhaps more emphasis should be given to the results giving useful indications, but that the issues should be looked at in detail for any specific case. (Line 109-110: yes, this approach makes the results less specific, but also less relevant to any particular case.)*
   **[Authors]** We have rewritten multiple paragraphs of the article to improve clarity and readability. The limitations of the study were already described in detail in multiple places in the original manuscript, and especially in the introduction and conclusions. These parts have now been further expanded. We believe that the revised version of the paper presents in a clear, transparent and objective way all assumptions and limits of the present study, putting the reader in a position to correctly interpret the conclusions.

2. **[Reviewer]** *Abstract wording, line 7: "with potentially benefits" suggest "with potential benefits", and in "essentially limited to the sole tower", suggest deleting "sole".*
   **[Authors]** Thank you for these corrections, we have modified the text.

3. **[Reviewer]** *Load cases, Table 1: What about other fault cases? Further down it seems they are assumed to be unaffected, but often they need very careful thought, and some may prove to be troublesome. As an example, what about the case of ultimate loads when the lidar fails to provide a good signal for any reason, and the failure is not detected? There will always be times when the lidar signal is unavailable, when the controller should revert to a safe mode. I don't think this has been considered. If the failure is undetected, so safe mode is not activated, it is possible that the control will result in higher ultimate loads than without any lidar at all. This is just one example.*
   **[Authors]** Thank you for pointing this out. As written in the text, this work considers a lidar availability of 100% and does not account for the effects of faults.
   Mitigating the effects of faults could be obtained through the adoption of a triple redundant system. Clearly, the adoption of such a system would require higher purchase and maintenance costs, which would reduce the economic benefits of the redesigns (considering the current technologies), as discussed in the conclusion section.
   Higher ultimate loads due to faulty conditions could be more problematic for the rotor than for the tower. The design of fatigue-driven towers will most likely not be affected significantly by higher ultimate loads, as these do not drive the design, and the associated constraints are satisfied with significant margins (as shown in Section 3.2). In contrast, for the rotor the

situation is more complex, as important driving constraints -such as tip deflection- are clearly dependent on ultimate conditions.

We have highlighted in different parts of the text these limits of our analysis and the need to include faulty situations in future studies. In addition, we have added in Sect. 2.3 a dedicated paragraph that discusses the problem of ultimate loads. As argued there (but also elsewhere in the text, and also in the previous version of the manuscript), the design of components that exploits LAC-induced reductions of design-driving ultimate loads is not a trivial task. Although this problem can only be addressed with reference to specific situations, and therefore falls outside of the scope and spirit of this work, having neglected it here does not modify the conclusions of the present analysis. In fact, for the cases considered here, only fatigue-related design-driving constraints could be relaxed thanks to DEL reductions, whereas reductions of ultimate loads did not have any impact on the final designs.

The discussion on this important point has been significantly expanded in the revised manuscript, and will hopefully not leave any doubts in the readers.

4. **[Reviewer]** *The results should include a strong caveat about load cases that have not been considered but that may affect the conclusions. (Line 139: "Hence, LAC-induced load reductions were assumed to be null for these DLCs, which is a conservative choice" - it is not necessarily conservative - LiDAR could make things worse in some situations.)*
**[Authors]** We agree, and we have removed the sentence. Additionally, as explained earlier on, the discussion on ultimate loads has been refined and greatly expanded.

5. **[Reviewer]** *It might be helpful to list the DLCs that are \*not\* considered, with reasons why not considered important (some will be obvious of course).*
**[Authors]** The complete list of DLCs included in the standards is shown as follows, with a description of the role of each DLC in the study:

| DLC | | | |
|---|---|---|---|
| | | Wind profile | **Role in the study** |
| **1. Power Production** | **1.1** | NTM | Considered, reduced by LAC load-reduction model |
| | **1.2** | NTM | Considered, reduced by LAC load-reduction model |
| | **1.3** | ETM | Considered, reduced by LAC load-reduction model |
| | **1.4** | ECD | Not considered, as not design driving |
| | **1.5** | EWS | Not considered, as not design driving |
| **2. Power production plus occurrence of fault** | **2.1** | NTM[1] | Considered, not possible to apply a LAC load-reduction model |
| | **2.2** | NTM[2] | Not considered, as not design driving |
| | **2.3** | EOG[3] | Considered, not possible to apply a LAC load-reduction model |
| | **2.4** | NTM[4] | Not considered, as not design driving |
| **3. Start up** | **3.1** | NWP | Not considered, as not design driving |

[1] Control system fault or loss of electrical network
[2] Protection system or preceding internal electrical fault
[3] External or internal electrical fault including loss of electrical network
[4] Control, protection or electrical system faults including loss of electrical network

| | 3.2 | EOG | Not considered, as not design driving |
|---|---|---|---|
| | 3.3 | EDC | Not considered, as not design driving |
| **4. Normal shut down** | 4.1 | NWP | Not considered, as not design driving |
| | 4.2 | EOG | Not considered, as not design driving |
| **5. Emergency shut down** | 5.1 | NTM | Not considered, as not design driving |
| **6. Parked (standing still or idling)** | 6.1 | EWM | Considered, not reduced by LAC load-reduction model (not dependent on controller) |
| | 6.2 | EWM | Considered, not reduced by LAC load-reduction model (not dependent on controller) |
| | 6.3 | EWM | Considered, not reduced by LAC load-reduction model (not dependent on controller) |
| | 6.4 | NTM | Not considered, as not design driving |
| **7. Parked and fault conditions** | 7.1 | EWM | Not considered, as not design driving |
| **8. Transport, assembly, maintenance and repair** | 8.1 | NTM | Not considered, as not design driving |
| | 8.2 | EWM | Not considered, as not design driving |

Therefore, as described in Section 3.2, the DLCs considered in this study are DLC 1.1, DLC 1.2, DLC 1.3, DLC 2.1, DLC 2.3, DLC 6.1, DLC 6.2 and DLC 6.3. From this list of DLCs, as described in Table 2, the LAC load-reduction model is applied to DLC 1.1, DLC 1.2 and DLC 1.3. DLC 2.1 and 2.3 are considered in the study but are not reduced with a LAC load-reduction model, as the effect of LAC can only be quantified in fully-coupled simulations. DLC 6.1, 6.2 and 6.3 are considered but not reduced, as they represent stand-still conditions that do not depend on the control strategy. The other DLCs are not considered in the study, since they are not driving the design of these specific three machines, as described in their respective reports (WT1[5], WT2[6] and WT3[7]).

We have modified several parts of the text to better explain which DLCs are considered and why.

6. **[Reviewer]** *Line 115: "guaranteeing" (spelling)*
   **[Authors]** Thank you, we have corrected this typo.

7. **[Reviewer]** *Line 169: "it is assumed that load reductions are independent of wind speed" – seems a strange statement. Load reductions may be much greater above rated when pitch control is active, and can depend strongly on wind speed.*
* * *
[5] Bak et al.: Description of the DTU 10 MW Reference Wind Turbine
[6] Bottasso C.L. et al.: Integrated aero-structural optimization of wind turbines. Multibody Syst. Dyn., 38, 317–344, https://doi.org/10.1007/s11044-015-9488-1, 2016
[7] Bortolotti P. et al.: IEA Wind TCP Task 37: Systems Engineering in Wind Energy – WP 2.1 Reference Wind Turbines Technical Report, https://doi.org/10.2172/1529216, 2019

**[Authors]** The assumption of load reductions being independent of wind speed is clearly an approximation, which only holds true when the reduction coefficients are small. The impact of this assumption was found to be negligible for small reduction coefficients.

This conclusion was obtained as follows. A dedicated study was conducted by first defining a wind-speed-dependent scenario, which was obtained from Bottasso et al. (2014)[8]. That study reports the wind-speed-dependency of the reduction in tower-base fore-aft DEL obtained with a Lidar-enhanced control strategy for a 3 MW machine. This wind-speed-dependent reduction scenario was applied to the WT3 machine, this way obtaining an equivalent Weibull-averaged DEL of -2.75%. A wind-independent scenario was then defined by assuming a constant profile, in which the reduction for each wind speed is equal to 2.75%, i.e. the same value as the Weibull-averaged DEL. The comparison between the two scenarios is shown in Fig. 1.

[Figure]

**Figure 1.** Comparison between the WS-Dependent (defined based on Bottasso et al. (2014)) and WS-Independent scenarios.

Finally, both scenarios were applied to the WT3 machine, and the fatigue margin at the tower base was computed, resulting in a change of only +1.02% for the WS-Independent scenario with respect to the (more accurate) WS-dependent one.

The study was extended by considering progressively larger reduction conditions for both the WS-Dependent and WS-Independent scenarios. Results shows that the difference between both scenarios increases for higher load reductions; however, the difference remains small. For example, considering the DEL flapwise blade root moment reduction of the model, which is equal to -3.8%, the difference in fatigue margin at the blade root between the WS-dependent and independent scenarios was found to be less than 2%; for the DEL fore-aft bending moment at tower base, which is equal to -12% according to the model, the fatigue margin difference was found to be approximatively equal to 5%. For all cases, the WS-Independent scenario leads to a higher fatigue margin, implying that the application of a WS-Independent scenario underestimates the effect of a real WS-Dependent scenario (and hence overestimates the benefits of LAC).
* * *
[8] Bottasso C.L. et al.: Lidar-enabled model predictive control of wind turbines with real-time capabilities.

For the load-reductions of the present investigation, the error generated by assuming the WS-independent scenario was assumed to be negligible, at the light of the other approximations already present in the analysis.

We have expanded the text in Sect. 2.3 to better explain this point of the analysis.

8.  **[Reviewer]** *Figure 6: Make figures bigger to help legibility. Why are some bars missing?*

   **[Authors]** We have increased the size of Figure 6 (Figure 7 in the revised version) to improve readability; however, the figure size will be eventually optimized by Copernicus during the production phase. The figures show the effects of LAC on the tower heights that were redesigned for each model. Indeed, the study is conducted for increasing heights until it is no longer economically feasible, or the upper boundary of 15% increment is reached. This means that the tower height of WT1 was increased until the upper margin of 15% was encountered, while the tower height of WT2 and WT3 were increased up to 5% and 10%, respectively, since no mass reduction was obtained at those heights.

   We have rewritten part of the text and the figure caption for better clarity.

9.  **[Reviewer]** *It would be nice to display the non-LAC mass increases compared to 20 yr life, and how they change with LAC.*

   **[Authors]** Figure 2 (in this reply to the reviewers) displays the mass of the updated non-LAC baselines (redesigned for each lifetime) with respect to the initial non-LAC 20-year baseline. Figure 3 reports the mass of the LAC-enabled redesigned towers with respect to the corresponding non-LAC baseline tower.

   These plots show how the LAC-enabled redesign counteracts very nicely the mass increase introduced by the longer lifetime.

   Figure 3 shows an unequal effect of LAC for the different lifetimes and models. Indeed, the impact of LAC on the redesigned structure can be limited by introducing geometric constraints that ensure a realistic shape.

[Figure]

**Figure 2.** Mass difference of the updated baseline towers (non-LAC) wrt initial 20-year baseline

[Figure]

**Figure 3.** Mass difference of the LAC-enabled redesigned tower wrt the updated non-LAC baseline

10. **[Reviewer]** *Line 429: "radar" should be "lidar"?*
    **[Authors]** Thank you for pointing this out. We have corrected this typo.

**Reviewer #2**

1. **[Reviewer]** *The paper deals with the benefits of including LAC in turbine design, evaluated in terms of LCOE. It is shown that LAC can reduce the mass of the tower, increase its height (which increases production) or extend its operating life. The LAC also allows to reduce the stress on the blades and thus to modify the structural properties of the rotor. The study is conducted considering the "modifiable" DELs, i.e. the DEL for which the LAC has an impact on the results. To evaluate the benefits of the LAC, the DELs and extreme loads are scaled-up, based on the results of a study Bossanyi, presented in 2014. A strong simplifying assumption is made regarding DEL scaling, which assumes that the reduction is independent of the wind speed and load range. This work seems to suggest that the gain obtained by the design of the turbines including the LAC is negligible in most cases, given the current costs of Lidar and its maintenance costs. The authors are aware of the limitations of the study, and it is explicitly stated that these results are strongly impacted by the its assumptions. Anyway, this issue (identified by the IEA Task 32) has a real interest, and this paper provides a good starting point for its understanding. In my opinion, an important improvement to this work would be to consider the variability of the gains given by the LAC (or even to determine from which gains LAC becomes economically interesting), as the potential benefits depend on the used control technique, on its settings, on the operating conditions and, obviously, on the NON-LAC technique to which LAC is compared. This would have the merit of identify a level of performance that LAC would have to reach in order to be economically interesting, and could strongly incentive further research efforts on this topic. Minor remark: At some points the paper is a bit hard to read. In this sense, a schematic summary or table, and an effort to make the explanations simpler and clearer can make the paper more readable.*

    **[Authors]** Thank you for your comments. We have rewritten several parts of the article and added a schematic plot (Figure 1 in paper) to improve readability and comprehension.

The reviewer's comment on wind speed independency is already address in reply no. 7 to reviewer #1.

We agree that it is necessary to extend the study to consider the variability of gains beyond the 3 scenarios analyzed here. The original text already pointed out clearly this and several other limitations of the study, but these parts have now been expanded further.

We have taken the opportunity to make several small editorial changes to the text, in order to improve readability. A revised version of the manuscript is attached to the present reply, with deletions marked in red and additions in blue.

The authors

---

## Author Response (AR2)

**REVISION TO MANUSCRIPT DRAFT**

**Wind Energy Science Discussion**

**What are the benefits of lidar-assisted control in the design of a wind turbine?**

Dear Editor-in-Chief Prof. Mann, thank you for your suggestion to improve fig. 10, which we have now updated according to your indications.

Best regards,

The authors